# Evaluation of the Friday Night Live Mentoring Program on Supporting Positive Youth Development Outcomes

**DOI:** 10.3390/healthcare12212199

**Published:** 2024-11-04

**Authors:** Kathleen P. Tebb, Ketan Tamirisa

**Affiliations:** 1Department of Pediatrics, Division of Adolescent and Young Adult, Medicine UCSF Benioff Children’s Hospital, University of California, 550 16th St. Fourth Floor, Box 0503, San Francisco, CA 94143, USA; 2Department of Public Health, Washington University in St. Louis, 1 Brookings Drive, St. Louis, MO 63130, USA; s.tamirisa@wustl.edu

**Keywords:** positive youth development, mentoring, adolescent health, youth health, ATOD, ATOD prevention

## Abstract

Introduction: The use of alcohol, tobacco, and other drugs (ATOD) is a leading cause of preventable morbidity and mortality among adolescents. While traditional interventions have targeted specific health-risk behaviors (e.g., substance use, initiation of sexual intercourse, truancy, etc.), the evidence suggests that using a positive youth development (PYD) framework may have positive impacts across a number of domains. Friday Night Live Mentoring (FNLM) is a PYD-based, cross-age peer mentoring program that engages teams of older high school-aged youth to mentor teams of middle school-aged youth in a structured, ongoing, one-on-one relationship. While studies have demonstrated significant but small effect sizes of intergenerational youth mentoring programs in which an adult mentor is paired with the youth mentee, research on cross-age mentoring programs is limited. The purpose of the current study is to evaluate FNLM on its ability to improve participants’ knowledge, attitudes, skills, opportunities to develop caring relationships, school engagement, and academic performance. Methods: A retrospective, pre–post survey was administered online to FNLM participants across 13 California counties. Participants rated their knowledge and attitudes about ATOD, skills, relationships with peers and adults, and academic indicators. Open-ended questions gathered information about participants’ experiences in FNLM. Non-parametric related-samples Wilcoxon signed rank tests (an alternative to paired *t*-test) were used to compare pre–post differences. Participants were also asked two open-ended questions: “What are the best parts of FNLM?” and “What, if anything, would you change?”. The responses to each question were reviewed, coded, and analyzed according to key themes. Results: A total of 512 participants completed the survey (287 mentors and 225 protégés). There were small but statistically significant improvements across all items for both mentors and protégés. Qualitative analyses showed that most mentors and protégés especially enjoyed getting to know and spend time with one another. Several mentors added that it was rewarding to be a positive influence on or to make a positive difference in the protégé’s life. Many youth stated that the relationships formed, especially with their partner, and the activities were the best part of FNLM. The overwhelming majority would not change anything about the program. Those who provided recommendations for program improvement suggested more activities or more hands-on and engaging activities and more or longer meetings. Conclusion: FNLM actively engages youth and provides them with support and opportunities that promote knowledge, skill development, positive relationships, academic engagement, and success and raise awareness of the harms that the use of alcohol, tobacco, and other drugs (ATOD) can cause. While ATOD use was low prior to program participation, it was significantly lower after participating in the program.

## 1. Background

### 1.1. Burden of the Problem

Adolescents’ use of alcohol, tobacco, and other drugs (ATOD) is a critical public health issue [1]. ATOD use is a leading cause of preventable morbidity and mortality among adolescents and increases the risk of academic, psycho-social, physical, mental, and sexual health problems as well as accidents and unintentional injuries and can cause detrimental changes in brain development [2,3,4]. Alcohol use and binge drinking among U.S. high school students remains high, and 17% report riding with a driver who had been drinking [5,6]. Use of alcohol at an early age is associated with alcohol-related problems in adulthood [7]. Among those who began drinking under 14 years of age, 45% developed a dependence on alcohol later in life, and if alcohol initiation was delayed to over 21 years, lifetime alcohol dependence dropped to 10% [8]. Similarly, tobacco use is common among youth, with about 19% of middle and high school students reporting ever having used any tobacco product, with the most prevalent forms being electronic cigarettes [9]. Tobacco use, including e-cigarettes, smokeless tobacco, or nicotine gummies or pouches, during early childhood and adolescence causes significant respiratory illnesses, decreased physical endurance, bronchitis, and reduced pulmonary function tests [10]. Moreover, it leads to cancer, cardiac and vascular disease, and stroke later in life and ultimately contributes to increased mortality [11]. Among adults who are addicted to nicotine, nearly 80% had smoked tobacco by the time they were 18 years of age, and nearly 95% had smoked a cigarette by age 21 [12]. The percentage of adolescents reporting other illicit substances was assessed in the Monitoring the Future Survey, and it found that nearly one in ten eighth graders, one in five tenth graders, and three in ten twelfth graders were using illicit drugs [13]. There are also disturbing trends of deaths due to overdose from other drugs (i.e., fentanyl and opioid use). Specifically, fentanyl-related deaths among adolescents were four times higher in 2021 compared to those in 2019 (1.21 per 100,000 in 2019 to 4.23 per 100,000 in 2021) [14].

### 1.2. Interventions

Given that most adults with substance abuse problems started using in their teens, and early adolescence is a key period for substance use experimentation, prevention efforts targeting middle and junior high school students are particularly crucial [8]. Research shows that the most effective prevention programs are based on positive youth development (PYD) and are grounded in psychosocial theories of substance use [15]. Effective PYD programs share several features, including an emphasis on building upon youths’ strengths and skills, fostering caring relationships, supporting meaningful youth engagement in developing and executing activities, and creating opportunities to make meaningful contributions [16]. PYD programs have been shown to protect youth against tobacco and alcohol initiation, promote social skills, reduce sexual risk, and increase self-sufficiency, responsibility, and civic participation [17,18,19,20,21]. As a result, the National Substance Abuse and Mental Health Services Administration (SAMHSA) developed a Strategic Prevention Framework (SPF), which recommends incorporating these PYD strategies into ATOD-prevention efforts [22].

Positive relationships with peers and adults are critical to adolescent development. These relationships contribute to social, emotional, and cognitive development and play a crucial role in transitioning from youth to adulthood [23]. Positive peer connections are strongly associated with adolescent well-being and school connectedness [24]. Similarly, adolescents with connections with adults experience better school performance and less involvement in violence and substance use [25,26,27]. Some prevention programs that incorporate interconnectedness with peers or adults have been shown to reduce overall ATOD use with improved youth well-being [28]. While research is limited, a meta-analysis of six cross-age peer mentoring programs (where a supportive relationship with an older youth mentor and a younger mentee of the same generation) improved school, social, health, cognitive, and psychological outcomes, especially when there is adult supervision [29].

### 1.3. FNLM Intervention

Friday Night Live Mentoring (FNLM) is a program that establishes peer-to-peer mentoring relationships by pairing high school-aged mentors with middle school-aged protégés. Principals and teachers identify middle school-aged youth who may be at risk of a difficult transition to high school and who might benefit from a high school mentor, and then, they recommend these middle schoolers to the program. School staff and the FNLM adult staff (advisors and coordinators) recruit high school students (grades 10–12) who they think will be good mentors. FNML staff are encouraged to consider recruiting youth who might not be involved in other activities but who have a lot to offer and to consider selecting mentors similar to the protégé population (e.g., gender, ethnicity, life interests, and whose strengths align with the needs of the protégés). Youth can also self-refer to participate. Youth who want to be a mentor must complete an application, have their parent’s/guardian’s consent, and participate in an oral interview with the advisor and program coordinator. Once they have been selected, they are required to sign a contract related to their commitment to the program. This includes their agreement to act as a role model at all times, make a commitment to spend up to three hours a week dedicated to the mentoring program, and maintain a grade point average (GPA) of 2.5. There must be a three-year grade difference between mentors and protégés. FNLM staff then match mentors with protégés. There are 16 sessions in the FNLM program, which cover topics such as decision-making, substance use awareness and prevention, peer influence, tobacco, and e-products. In the 2023–2024 program year, 13 counties in California implemented the FNLM program. FNLM is based on evidence-informed standards of practice that include providing a physically and emotionally safe environment, opportunities to develop caring and meaningful relationships with adults and peers, opportunities for involvement and connection to community and school, opportunities for leadership and advocacy, and opportunities to engage in skill-building activities.

### 1.4. Purpose of Our Study

The purpose of the current study is to evaluate FNLM on its ability to improve participants’ knowledge, attitudes, skills, opportunities to develop caring relationships, school engagement, and academic performance.

## 2. Methods

A retrospective, pre–post survey was administered online to FNLM participants between 11 March and 13 May 2024 to evaluate the FNLM program for the 2023–2024 program year. Participants were asked to answer each question “before” they joined FNLM and “now” after program participation using a 4-point rating scale to assess knowledge, skills, relationships with peers and adults, commitment to school, perceptions of ATOD harm, and commitment not to use ATOD. Grades were assessed with a 5-point scale. The survey also included five open-ended questions to gather in-depth information about the youths’ experiences in FNLM from both the perspectives of the mentors and protégés.

This survey was first developed in 1996 and went through a scientific validation process in 2004 to capture accurate and reliable evidence to assess if participation in the FNLM program contributes to meaningful change. A retrospective pre–post survey was chosen because of its advantages over traditional pre–posttest surveys, including implementation feasibility (administered at just one time-point), assurance that there are paired data for each participant, and reduced measurement error through response-shift bias [30,31]. Meaningful comparisons require participants to be able to use the same frame of reference to provide comparable self-assessments. A participant may not have sufficient information prior to program participation to accurately assess their baseline functioning, and participating in the program can shift their frame of reference used to answer a pretest question (response-shift bias), or they can over- or underestimate pretest assessments based on limited pre-intervention knowledge. There are limitations to this approach. It requires recalling attitudes, knowledge, and behaviors prior to their experience in the intervention [31]. This may be difficult due to time that has passed, their experience in the program, and memory-related challenges. Participants may also exhibit personal bias, as it is common for individuals to want to improve their knowledge and skills. Finally, while it ensures paired data, it does not capture information from youth who started but did not complete the program. Both approaches are subject to social desirability (desire to please program staff or make the program look good) [32]. To help ensure the accuracy of participants’ responses and to reduce social desirability, participation was voluntary and anonymous, and youth were encouraged to respond as honestly as possible. The survey was administered online to further assure the anonymity of responses.

The data were analyzed by an independent evaluator. Descriptive statistics were used to analyze demographic data, and a non-parametric paired-sample statistical test, the Wilcoxon signed-rank sum, was used to test if the pre-post differences in the matched data were statistically significant. All analyses were conducted with SPSS version 29 (IMB, Chicago, IL, USA). Results were tabulated under the thematic categories of ‘knowledge’, ‘skills’, ‘making a positive difference in school/community’, ‘relationship with peers and adults’, ‘commitment to school and hope for the future’, ‘use of tobacco, alcohol, and other drugs’, ‘perceptions of harm of peoples’, ‘choices’, and ‘commitment to not use’. Responses to open-ended questions were coded and analyzed by frequency of themes. A coding guide was developed based on an initial review of responses. Two researchers independently coded 10% of the responses in each category with an inter-coder reliability >80%. Discrepancies were resolved, and coding guidance was further clarified.

A total of 512 youth responded to the survey (287 mentors and 225 protégés). The majority of participants identified as female, were from racially/ethnically diverse backgrounds, and few (36% of mentors and 27% of protégés) parents/guardians received a college degree (associate degree or higher) (Table 1).

## 3. Results

There were small but statistically significant pre–post improvements for each of the items assessed in the following areas: knowledge about problems in their community about and acceptance of ATOD and what to say when offered ATOD; a wide range of skills; relationships formed with peers and adults; academic indicators including attitudes toward school, grades, and attendance; hope for the future; perceptions of the harms that ATOD use can cause; commitment not to use ATOD; and use of ATOD. See Table 2 for the pre–post means for each item. There were no significant differences in ratings by demographic factors.

Most youth answered the open-ended questions (87% of the mentors and 75% of the protégés). In response to the question, “What are the best parts of FNLM?”, over half of the mentors (approximately 51%) said they enjoyed the people, especially getting to know and spend time with their protégés, and 17% stated they especially enjoyed being a positive influence on or making a positive difference in the protégé’s life. The following quotes reflect this theme:


*The best parts are getting to talk to new people and having time to get to know your protégé.*



*The friendships between mentors and protégés.*



*Spending time with the protégés and getting to know them.*



*You get to talk to younger students and influence them to keep going and don’t give up.*



*The best part was working with my protégé. Having the ability to help them with issues and feeling like I was having some influence on them.*



*I love being able to share my knowledge while also learning more about good choices I can make in different aspects of my life.*



*I love the fact that most mentors join together to help each other out; whether it’s to help the protégés out or with lesson plans, everyone has each other’s backs.*


Many of the mentors (34%) stated that the activities were the best part of FNLM, as they made the program fun and engaging, provided them with opportunities to bond with others, especially their protégés, and increased their knowledge and skills. The following quotes illustrate this theme:


*Interacting with my classmates and learning new things such as life skills and everyday things.*



*Meeting new people and developing leadership/group work skills.*



*Meeting people, shadow day, and the fun games. I really enjoy learning about such significant topics in a really fun way.*



*One-on-one time with protégées and playing fun games with them.*



*Playing games while learning.*



*The best parts are group work, socialization, and engaging activities that teach valuable information and skills.*


Some of the mentors (almost 6%) stated the best part was being more involved in their school or community, as indicated in the following quotes:


*I loved the planning and being more involved in my school.*



*I think the best part of mentoring FNL would be being involved with community activities.*



*Being involved.*



*Getting experience in helping out for educational events.*



*The best part of this experience would be getting to help out with the volunteering stuff.*



*Working with others and being involved.*


A few of the mentors (4%) said they liked “everything” about the program, and a small percentage (3%) reported that FNLM provided a positive environment and gave them a sense of community. Only three (1.2%) responded with “I don’t know”.

Similar to the mentors, over half of the protégés (59%) said they enjoyed the people, especially getting to know and spend time with their mentor. The following quotes represent this theme:


*I enjoyed talking to the mentors the most. Being able to socialize with others who have experienced things I have not yet experienced (such as high school) is an amazing experience.*



*I loved talking to my high school friends.*



*Making new friends.*



*Spending time with friends and learning from others. Feeling like the teenagers care about the middle schoolers.*



*I enjoyed the people because they were funny and helpful.*



*Getting to know the mentors more and more people who go to my school.*



*Getting to meet/see people that I get to connect with and hang out with.*



*Thank you for letting me meet with the mentors.*


Many protégés (26%) stated that the activities were the best part of FNLM. This is reflected in the following quotes:


*Group activities (things that include everyone together).*



*I enjoy doing activities with my friends and learning more things.*



*I enjoy doing the blindfold activity because I like doing outside activities and working with my mentor.*



*I enjoy playing the games with the mentors because they are fun and there is always an educational reason to play the game.*



*I enjoy the games and the things we learn in them.*



*[The] outside activities because it’s fun and you can interact with the mentors.*



*The helpful and learning games that we play.*


Some of the protégés (12%) reported that the best part was the knowledge and skills they gained in the program. A few (6%) said the food and snacks were the best part, while others (5%) stated they liked everything about the program. A small percentage (1.8%) stated it was fun but did not provide additional information, and another 1.8% responded with “I don’t know”.

In response to the question, “What, if anything, would you change about FNLM?”, many mentors (28%) and most of the protégés (59%) said they would not change anything about the program because they valued the program. Many liked the program so much that they wanted more or longer meetings, especially to spend more time with their mentor or protégé. The most common suggestion was to have more or different activities, especially to have them be more interactive or hands-on. This information was reported back to each program to inform their ongoing site-specific quality improvement efforts. The summary of the pre–post protégé ratings are listed in Table 3.

## 4. Discussion

A retrospective pre–post survey was a feasible approach to evaluate FNLM in the “real-world setting” and captured the perspectives of both mentors and protégés. Most youth responded to the survey, and analyses revealed statistically significant improvements in knowledge, skills, relationships, academic indicators, and perceptions of the harms of ATOD use. There were also significant reductions in the use of ATOD. The qualitative data showed that participants especially valued the relationships formed in the program, and they enjoyed the activities that supported positive relationships and increased program engagement and the participants’ knowledge and skills. There is a vast body of literature showing that connecting youth with caring adult mentors supports positive outcomes [33,34]; however, there is much less research on the effectiveness of peer mentors. Studies from across the globe have found that peer-led substance use prevention programs are an effective approach to reducing substance use, especially when there is careful attention to the selection and training of peer leaders [35,36,37]. While peer-led programs are distinct from 1:1 mentoring programs, peer leaders do serve as role models who are effective at changing social norms, providing alternatives to substance use, and teaching ATOD-refusal skills. Additional research shows that a strong positive peer network contributes to improved social and academic outcomes [38,39,40]. The current study contributes to a growing body of research demonstrating that peer mentoring of high school students with younger students benefits both the protégé and the mentor. [41,42]. 

### Limitations

There are some noteworthy study limitations. FNLM is a voluntary program; thus, youth who participate may be more amenable to the potential impacts of the intervention, and findings may not be generalizable to all youth. While our study included a diverse population of adolescents, there are some subgroups within the race/ethnicity and gender identity categories that contain small sample sizes, which limits generalizability to these groups of adolescents. Other extraneous factors can also affect the outcomes further limiting generalizability (such as characteristics of the school site or staffing, experiences with other programs/activities, or participant characteristics not measured in this study). While this was a racially/ethnically diverse and gender-diverse sample, some subgroups had few participants, which limits generalizability to different populations of adolescents. The study was further limited because it was not feasible to include a comparison group. Therefore, it is not possible to definitively conclude that improvements were a direct result of the intervention, and it is possible that youth could have developed positive outcomes without participating in FNLM. Furthermore, we were not able to match mentors and protégés by 1:1 correspondence. While this is a limitation, it allowed us to make sure that the survey was anonymous, especially since we were asking about participants’ specific risk behaviors.

There are also limitations inherent in a retrospective pre–post design. Specifically, while a retrospective survey captures complete pre–post responses from participants, it lacks information for youth who started the program but did not complete it. Additionally, because participants are asked to recall information about how they felt before the intervention and how those perceptions have changed, results may not be completely accurate due to variability in establishing a clear “before state” and reliance on autobiographical memory [31,43].

Despite the limitations to this approach, there are also some advantages to this design. Compared to traditional pre–post surveys (conducted at two-time points prior to the intervention and after the intervention), retrospective pre–post surveys have been found to provide a more accurate assessment of participants’ perception of change because both answers are generated within the same frame of reference, allowing participants to more accurately assess their functional baseline [44]. This retrospective pre–post design also helps mitigate response-shift bias, where participants possess more positive attitudes before an intervention compared to after, which is inherent in most traditional pre–post-designed studies. Further, it can reduce the influence of other outside events on participants’ responses that might take place during the time lapse between a traditional pre–post assessment [30,44]. Thus, confounding variables and factors are mitigated using this design. In addition, compared to the traditional pre–post survey, the retrospective pre–post has been found to be no more susceptible to social desirability or compliance with implicit task demands [45]. This study also used a retrospective pre–post survey design that captured self-reported data. To the extent participants were trying to improve their skills and wanted to see improvements, responses could be subject to bias [32,45]. With any survey that captures self-reported data, social desirability and motivation for making the program look good is a concern; however, administration procedures were implemented to reduce this potential bias.

## 5. Conclusions

FNLM is an evidence-informed, positive youth development program that builds supportive, caring relationships between adults, mentors, and protégés to support the development of protective factors that contribute to youth resilience and positive outcomes [8,9,10,11,12,13]. Despite any limitations, FNLM is a promising intervention for reducing ATOD and promoting positive youth development outcomes. This current evaluation of FNLM shows that youth experienced significant improvements across several domains: knowledge of ATOD, skills, caring relationships, school engagement, perceptions of ATOD harm, and commitment to avoid ATOD. The combined quantitative and qualitative data further suggest that FNLM actively engages youth and provides them with support and opportunities that promote skill development, social competencies, mental health, and academic success.

### Future Implications

The results of this FNLM program are promising, revealing that PYD programs are effective in helping to advance youth health and prevent AOTD use. Incorporating PYD programs into prevention efforts will continue to have far-reaching implications, allowing adolescents, especially those vulnerable and in underserved communities, to experience holistic improvements in development and overall health. Thus, alongside targeted intervention strategies aimed at reducing AOTD use and promoting healthy behaviors, PYD programs similar to this one should be implemented at schools and community-based organizations. Above all, it is important to prioritize establishing education and comprehensive early intervention efforts like this program to mitigate health-risk behaviors among youth and improve outcomes.

## Figures and Tables

**Table 1 healthcare-12-02199-t001:** Participant Demographics.

Variable	Mentors N (%)		Protégés N (%)	
**Gender**				
Female	176	(61.3)	100	(44.4)
Male	69	(24.0)	82	(36.4)
Gender fluid	3	(1.0)	4	(1.8)
Gender non-binary	6	(2.1)	3	(1.3)
Transgender	2	(0.7)	3	(1.3)
Two-Spirit	1	(0.3)	0	(0.0)
Missing/Unsure/Not Stated	30	(10.5)	33	(14.7)
**Race/Ethnicity ***				
African American/Black	19	(5.8)	20	(8.5)
Asian/Pacific Islander	48	(14.5)	29	(12.3)
Middle Eastern/North African	8	(2.4)	7	(3.0)
Latino/Hispanic	143	(43.3)	72	(30.6)
Native/Indigenous	14	(4.2)	14	(6.0)
White/European	91	(27.6)	77	(32.8)
Not listed/Declined to state	7	(2.1)	16	(6.8)
**Primary language spoken at home**				
English only	104	(36.2)	98	(43.6)
Language in addition to or other than English **	159	(55.4)	87	(38.7)
Missing/Did not report	24	(8.4)	40	(17.8)
**Length of involvement in FNLM**				
Less than one semester	36	(14.0)	44	(22.0)
One semester	55	(21.0)	49	(25.0)
Two semesters	106	(40.0)	61	(31.0)
More than one school year	65	(25.0)	44	(22.0)
Missing/Did not report	25	(8.7)	27	(12.0)

* Participants were given the option to select more than one race/ethnicity. ** Commonly spoken languages included Spanish, Arabic, and Chinese/Cantonese.

**Table 2 healthcare-12-02199-t002:** Pre–Post Difference in Mentor Ratings.

Items	Mentors	
	Pre Mean (SD)	Post Mean (SD)
**Knowledge** (For each statement, select how much you knew about the topic before you joined FNLM and how much you know about it now.) *Rating Scale: 1 = Nothing, 2 = A little, 3 = Somewhat, 4 = Well-informed* The problems in my community related to alcohol, tobacco, and marijuana use	2.79 (0.95)	3.35 (0.95) ^+^
My community’s attitudes and acceptance related to alcohol, tobacco, and marijuana	2.61 (0.94)	3.23 (0.96) ^+^
What to do or say when someone offers me alcohol, marijuana, or other drugs	3.05 (1.16)	3.43 (1.05) ^+^
**Skills** (For each statement, select how confident you were before you joined FNLM and how confident you are now.) *Rating Scale: 1 = Not at all, 2 = A little, 3 = Somewhat, 4 = Very*		
Working as part of a group.	3.03 (0.85)	3.74 (0.58) ^+^
Leading group discussions or meetings	2.51 (0.90)	3.35 (0.78) ^+^
Public speaking	2.62 (0.92)	3.36 (0.75) ^+^
Considering other group members’ points of view and opinions	3.10 (0.81)	3.68 (0.62) ^+^
Planning and organizing my time	2.80 (0.92)	3.41 (0.77) ^+^
Actively listening (i.e., carefully listening and showing the other person that you understand what s/he is saying)	3.19 (0.83)	3.72 (0.59) ^+^
Standing up for myself	2.97 (0.96)	3.44 (0.71) ^+^
Standing up for other people	3.00 (0.90)	3.47 (0.71) ^+^
**Making a Positive Difference in School/Community** (For each statement, select how you felt about each area before joining FNLM and how you feel now.) *Rating Scale: Rating Scale: 1 = Not at all 2 = A little, 3 = Somewhat, 4 = A lot* I know how to make my school a more positive place for everyone.	2.86 (0.88)	3.55 (0.73) ^+^
I know how to influence my peers to make positive choices around alcohol, tobacco, marijuana, and other drugs.	2.80 (0.94)	3.51 (0.80) ^+^
I work to make things better in my community.	2.76 (0.89)	3.40 (0.74) ^+^
**Relationships with Peers and Adults** (For each statement, select how you felt about each area before you joined FNLM and how you feel now.) *Rating Scale: 1 = Not at all, 2 = A little, 3 = Somewhat, 4 = A lot* I have worked with and/or learned about youth who come from different backgrounds (racial/ethnic, religious, economic, gender, or sexual identity)	2.88 (0.91)	3.53 (0.70) ^+^
I can think of people that I look up to.	2.99 (0.88)	3.46 (0.74) ^+^
I spend time with adults in a positive way.	3.08 (0.83)	3.58 (0.66) ^+^
There are adults who care about me.	3.39 (0.83)	3.66 (0.64) ^+^
There are adults in my life who believe I will be a success.	3.38 (0.80)	3.67 (0.62) ^+^
I have opportunities to make new friends.	2.96 (0.95)	3.55 (0.69) ^+^
**Commitment to School and Hope for the Future** (For each statement, select how you felt about each area before joining FNLM and how you feel now.) *Rating Scale: 1 = Not at all, 2 = A little, 3 = Somewhat, 4 = A lot* I am more excited about going to school.	2.38 (0.90)	2.89 (0.90) ^+^
I feel like there are people at my school who care about me.	2.88 (0.87)	3.33 (0.77) ^+^
I am committed to doing well in school.	3.33 (0.85)	3.62 (0.64) ^+^
I am more likely to continue my education (either through college or specialized training).	3.52 (0.76)	3.76 (0.57) ^+^
I feel like I belong/am more a part of my school.	2.74 (0.85)	3.40 (0.75) ^+^
I feel hopeful for my future.	3.22 (.88)	3.57 (0.69) ^+^
What grades did you get in school before you joined FNLM, and what grades do you get now?	3.25 (1.00)	3.53 (0.69) ^+^
About how many times did you skip school or cut classes each month before you joined FNLM and now?	0.83 (1.43)	0.53 (1.15) ^+^
**Use of Tobacco, Alcohol, and Other Drugs** *Rating Scale: 1 = Never, 2 = Hardly ever, 3 = Sometimes, 4 = Often* How often did you use tobacco before you joined FNLM, and how often do you use it now?	1.04 (0.28)	1.02 (0.21) ^v^
How often did you use alcohol before you joined FNLM, and how often do you use it now?	1.17 (0.51)	1.09 (0.34) ^+^
How often did you use marijuana before you joined FNLM, and how often do you use it now?	1.12 (0.39)	1.05 (0.30) ^+^
**Perceptions of Harm of People’s Choices** (How much do you think people risk harming themselves physically or in other ways when they engage in the following behaviors before you joined FNLM, and how much do you think they risk harming themselves now (after you participated in FNLM)?) *Rating Scale: 1 = No risk, 2 = Slight risk, 3 = Moderate risk, 4 = A great risk of harm*		
When they smoke a few cigarettes every day.	3.39 (0.89)	3.64 (0.81) ^+^
When they smoke cigarettes some days but not every day.	3.03 (0.91)	3.39 (0.86) ^+^
When they use marijuana sometimes (once a month or less).	2.54 (1.04)	2.95 (0.99) ^+^
When they use marijuana twice a week or more.	3.00 (1.00)	3.35 (0.95) ^+^
When they drink alcohol sometimes (three alcoholic beverages a month or less).	2.60 (1.00)	2.92 (0.98) ^+^
When they have five or more alcoholic beverages once or twice a week.	3.27 (0.94)	3.56 (0.86) ^+^
**Commitment to Not Use Substances** (Before you joined FNLM, how hard was it for you to say “no” to a friend who offered you the following substances? How difficult would it be for you to say “no” now?) *Rating Scale: 1 = Very hard, 2 = Hard, 3 = Easy, 4 = Very easy* If a friend offers you tobacco?	3.60 (0.87)	3.71 (0.80) ^+^
If a friend offers you alcohol?	3.45 (0.92)	3.66 (0.81) ^+^
If a friend offers you marijuana?	3.53 (0.89)	3.71 (0.77) ^+^

^+^ *p* ≤ 0.001; ^v^ *p* ≤ 0.05.

**Table 3 healthcare-12-02199-t003:** Pre–Post Difference in Protégé Ratings.

Items	Protégés	
	Pre Mean (SD)	Post Mean (SD)
**Knoweldge** (For each statement, select how much you knew about the topic before you joined FNLM and how much you know about it now.) *Rating Scale: 1 = Nothing, 2 = A little, 3 = Somewhat, 4 = Well-informed* The problems in my community related to alcohol, tobacco, and marijuana use.	2.56 (1.01)	3.28 (1.08) ^+^
My community’s attitudes and acceptance related to alcohol, tobacco, and marijuana.	2.26 (0.98)	2.97 (1.12) ^+^
What to do or say when someone offers me alcohol, marijuana, or other drugs.	2.96 (1.16)	3.43 (1.06) ^+^
**Skills** (For each statement, select how confident you were before you joined FNLM and how confident you are now.) *Rating Scale: 1 = Not at all, 2 = A little, 3 = Somewhat, 4 = Very*		
Working as part of a group.	2.81 (0.91)	3.60 (0.67) ^+^
I learn how to work with people that I don’t always agree with.	2.51 (1.01)	3.08 (0.95) ^+^
Speaking in front of others.	2.54 (0.97)	3.27 (0.88) ^+^
Considering other group members’ points of views and opinions.	2.84 (0.85)	3.57 (0.67) ^+^
Planning and organizing my time.	2.51 (0.96)	3.21 (0.90) ^+^
Actively listening: carefully listening and showing the other person that you understand what s/he is saying.	2.92 (0.94)	3.60 (0.70) ^+^
Standing up for myself.	2.71 (1.00)	3.40 (0.82) ^+^
Standing up for other people.	2.86 (1.00)	3.43 (0.82) ^+^
**Making a Positive Difference in School/Community** (For each statement, select how you felt about each area before joining FNLM and how you feel now.) *Rating Scale: 1 = Nothing 2 = A little, 3 = Somewhat, 4 = A lot* How to make my school a more positive place for everyone.	2.61 (0.99)	3.37 (0.82) ^+^
How to influence my peers to make positive choices around alcohol, tobacco, marijuana, and other drugs.	2.61 (1.03)	3.40 (0.91) ^+^
We try to make things better in the community or school.	2.56 (0.95)	3.35 (0.84) ^+^
**Relationships with Peers and Adults** (For each statement, select how you felt about each area before you joined FNLM and how you feel now.) *Rating Scale: 1 = Not at all, 2 = A little, 3 = Somewhat, 4 = A lot* I get to spend time with young people who are different from me.	2.57 (1.00)	3.33 (0.84) ^+^
I can think of people that I look up to.	2.79 (1.00)	3.39 (0.87) ^+^
I have opportunities to spend time with adults in a positive way.	2.78 (0.95)	3.48 (0.80) ^+^
There are adults who care about me.	3.16 (0.97)	3.58 (0.80) ^+^
There are adults in my life that believe I will be a success.	3.12 (1.00)	3.59 (0.77) ^+^
I have opportunities to make new friends.	2.84 (0.94)	3.47 (0.76) ^+^
**Commitment to School** (For each statement, select how you felt about each area before joining FNLM and how you feel now.) *Rating Scale: 1 = Not at all, 2 = A little, 3 = Somewhat, 4 = A lot* I am more excited about going to school.	2.13 (1.01)	2.85 (1.10) ^+^
I feel like there are people at my school who care about me.	2.72 (0.97)	3.23 (0.85) ^+^
I am committed to doing well in school.	3.05 (1.00)	3.45 (0.80) ^+^
I feel more prepared for high school.	3.04 (1.11)	3.50 (0.82) ^+^
I plan to go to college or another form of training after I graduate high school.	3.52 (0.76)	3.76 (0.57) ^+^
What grades did you get in school before you joined FNLM, and what grades do you get now?	2.88 (1.18)	3.14 (1.05) ^+^
About how many times did you skip school or cut classes each month before you joined FNLM, and how often do you skip school or cut classes each month now?	0.71 (1.41)	0.29 (0.87) ^+^
**Use of Tobacco, Alcohol, and Other Drugs** *Rating Scale: 1 = Never, 2 = Hardly ever, 3 = Sometimes, 4 = Often* How often did you use the following tobacco before you joined FNLM, and how often do you use it now?	1.09 (0.41)	1.02 (0.23) ^v^
How often did you use the following alcohol before you joined FNLM, and how often do you use it now?	1.15 (0.52)	1.04 (0.26) ^+^
How often did you use the following marijuana before you joined FNLM, and how often do you use it now?	1.15 (0.55)	1.04 (0.29) ^+^
**Perceptions of Harm of People’s Choices** (How much do you think people risk harming themselves physically or in other ways when they engage in the following behaviors before you joined FNLM, and how much do you think they risk harming themselves now (after you participated in FNLM)?) *Rating Scale: 1 = No risk of harm, 2 = Slight, 3 = Moderate, 4 = A great risk of harm* When they smoke a few cigarettes sometimes (about one cigarette a day or less).	2.58 (1.00)	3.17 (1.02) ^+^
When they smoke one or more packs of cigarettes a day.	3.13 (1.06)	3.55 (0.92) ^+^
When they use marijuana sometimes (once a month or less).	2.43 (1.08)	3.07 (1.05) ^+^
When they use marijuana twice a week or more.	2.83 (1.10)	3.34 (1.03) ^+^
When they drink alcohol sometimes (three alcoholic beverages a month or less).	2.30 (1.07)	2.92 (1.10) ^+^
When they have five or more alcoholic beverages once or twice a week.	2.93 (1.08)	3.39 (1.02) ^+^
**Commitment to not Use Substances** (Before you joined FNLM, how hard was it for you to say “no” to a friend who offered you the following substances? How difficult would it be for you to say “no” now?) *Rating Scale: 1 = Very hard, 2 = Hard, 3 = Easy, 4 = Very easy* If a friend offers you tobacco?	3.41 (0.96)	3.66 (0.80) ^+^
If a friend offers you alcohol?	3.39 (0.98)	3.64 (0.79) ^+^
If a friend offers you marijuana?	3.40 (1.03)	3.67 (0.77) ^+^

^+^ *p* ≤ 0.001; ^v^ *p* ≤ 0.05.

## Data Availability

The data that support the findings of this study are available from the primary author, [K Tebb], upon reasonable request.

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
