# Peer review of "Evaluation of the Friday Night Live Mentoring Program on Supporting Positive Youth Development Outcomes"

_healthcare, 2024, doi:10.3390/healthcare12212199_

Round 1
Reviewer 1 Report
Comments and Suggestions for Authors
Review: Journal of Healthcare – (10/10/2024)
Manuscript Number: healthcare-3251703
Manuscript Title: Evaluation of the Friday Night Live Mentoring Program on Supporting Positive Youth Development Outcomes
Overall, the manuscript was easy to follow and did not contain jargon that made it difficult to understand. The discussion section was easy to follow by using the purpose section as a guide for what to look for changes in.
Although the qualitative analysis was in between the tables, for ease of understanding of the reader, however, it is unclear if this manuscript was written with any specific writing style in mind. If APA style, tables need to be at the end of the manuscript, not listed within the text. Additionally, It would be nice to provide the explanation of the acronym ATOD in the background at least once as I found myself forgetting what it stood for while reading. Just a refresher occasionally in various sections would help. Finally, it will strengthen the results if you were to add information related to the extraneous variables mentioned that could have also affected the data. Since the students were not immersed only in the program, there could have been other factors like the school, etc that may not be generalizable. Please see the below feedback:
· Results
o (Page 4, paragraph 3, & line 176-180)
â–ª Feedback: Moving the demographics to the methods section would help the reader know what the demographics were before getting to the results.
o (Page 8 , in between lines 290-291, Table 3.)
â–ª Feedback: This may be a mistake, but for table 3, it says “Pre-Post Difference in Protégé ratings”, but the table still says mentors above the pre mean and post mean.
· References
o (Page 12-13, line 384-464)
â–ª References should be listed and formatted according to a specific writing style.
Summary
· This manuscript is a great fit for the journal. I really liked reading the article, and I can’t wait to see what the finished product is! Its easiness to read caused me to be invested in this intervention, and I hope it goes national. Others need to see the effects!

Author Response
Comment 1: If APA style, tables need to be at the end of the manuscript, not listed within the text.
Thank you for this suggestion. We have moved the tables to the end of the manuscript. We defer to the editor on the final placement of the tables.
Comment 2: Additionally, It would be nice to provide the explanation of the acronym ATOD in the background at least once as I found myself forgetting what it stood for while reading.
Thank you for this insightful comment. We have added a description of ATOD in the background to make it more clear to our readers.
Comment 3: Finally, it will strengthen the results if you were to add information related to the extraneous variables mentioned that could have also affected the data. Since the students were not immersed only in the program, there could have been other factors like the school, etc that may not be generalizable. Please see the below feedback:
Thank you. We have expanded the section on extraneous factors in the discussion section.
“Other extraneous factors can also affect the outcomes further limiting generalizability (such as characteristics of the school site or staffing, experiences with other programs/activities, or participant characteristics not measured in this study). While this was a racially/ethnically and gender diverse sample, some subgroups had few participants which limits generalizability to different populations of adolescents.”
Comment 4: Results
o (Page 4, paragraph 3, & line 176-180)
Feedback: Moving the demographics to the methods section would help the reader know what the demographics were before getting to the results.
Thank you. We included this part at the end of the methods section:
“A total of 512 youth responded to the survey (287 mentors and 225 protégés). The majority of participants identified as female, were from racially/ethnically diverse backgrounds, and few (36% of mentors and 27% of protégés) parent/guardian received a college degree (associate degree or higher)”
Comment 5:
o (Page 8 , in between lines 290-291, Table 3.)
Feedback: This may be a mistake, but for table 3, it says “Pre-Post Difference in Protégé ratings”, but the table still says mentors above the pre mean and post mean.
Thank you for pointing this out. We have corrected it the heading of the table.
Comment 6: References
o (Page 12-13, line 384-464)
References should be listed and formatted according to a specific writing style.
Thank you. We have formatted the references in APA style. Please note that there are some inconsistencies in the instructions for authors regarding referencing and the guidelines do not seem to match the style in published manuscripts for this journal. We defer to the editor on final formating of the references.
Reviewer 2 Report
Comments and Suggestions for Authors
This paper reports the results of an evaluation study of a cross-age peer mentoring program conducting among middle school-aged youth with the participation of high school-aged students. Intervention programmes are of great importance in reducing substance use among adolescents and it is important to evaluate these programmes. Therefore, the task of the paper is of central interest.
The background is well presented, and the literature review is relevant.
The method used by the authors is a retrospective pre-post survey. Using a retrospective method is an acceptable way of interviewing people, but it might have an effect on the results: people might have a different opinion when they look back than if they had been interviewed at the beginning of the intervention. I miss the discussion about the effect of retrospective questioning on the results. We can read about it in the 'limitations' section, but it should also be discussed in the methodology chapter.
The results are presented in tabular form and by quoting open-ended questions. I miss the interpretation of the figures in the tables. It would have been helpful to give a basic interpretation of the data and the results of the statistical tests. The title of table three reads: "Pre-post Difference in Protégé Ratings" while the heading of the table is "Mentors" - what this is about needs to be clarified. I suggest presenting the original questions in the tables to make them more understandable (e.g. not "perception of harm of people's choices" but what was the specific question).
Quotations are pleasant to read.
Discussion is very short. Here the author should give a more detailed interpretation of their results with some reflection on the literature.
Author Response
Comment 1: Authors appropriately discuss the limitations of the retrospective design used in the intervention. Of note too is the racial/ethnic profile of the sample. How may this limit generalizability of the findings to a wider youth population? Provide further details on how the youth were selected for the intervention.
Thank you for your comment on our discussiong o the retrospecitve design. We are also grateful for your addtional insightful comments. We have expanded our methods and discussion sections. Specifically in the discussion, we caution the reader about the generalizability of study findings given that there some of the racial/ethnic and gender identity subgroups have small samples. Also, we provided futher details about how youth were selected for this intervention in the background:
“School staff and the FNLM adult staff (advisors and coordinators) recruit high school students (grades 10-12) who they think will be a good mentor. FNML staff are encouraged to consider recruiting youth who might not be involved in other activities but who have a lot to offer, and consider selecting mentors similare to the protégé population (e.g., gender, ethnicity, life interests, and whose strengths align with the needs ot the protégés). Youth can also self-refer to participate. Youth who want to be a mentor must complete an application, have parent’s/guardian’s consent, and participate in an oral interview with the advisor and program coordinator. Once they have been selected, they are required to sign a contract related to their commitment to the program. This includes their agreement to act as a role model at all times, make a commitment to spend at up to three hours a week dedicated to the mentoring program, and maintain a grade point average (GPA) of 2.5. There must be a three-year grade difference between mentors and protégés.”
Comment 2: However, given the purpose of the study - which was to evaluate the FNLM outcomes, the discussion should have included a comparison/contrast with other such interventions
Thank you for this comment. We have expanded the discussion to address this issue and have included current literature for comparative purposes (along with citations).
“There is a vast body of literature showing that connecting youth with caring adult mentors supports positive outcomes [34-35]; however, there is much less research on the effectiveness of peer mentors. Studies, from across the globe, have found that peer-led substance use prevention programs are an effective approach to reducing substance use especially when there is careful attention to the selection and training of peer leaders [36-38]. While they are not 1:1 mentors, peer leaders serve as role models who are effective at changing social norms, providing alternatives to substance use, and in teaching ATOD refusal skills. Additional research shows that a strong positive peer network contributes to improved social and academic outcomes [39]. The current study contributes to a growing body of research demonstrating that peer mentoring of high school students with younger students benefits both the protégé [40] and the mentor [41].”
Thank you again for this constructive review as this and comments from the other reviewers helped improve our manuscript.

Reviewer 3 Report
Comments and Suggestions for Authors
Authors appropriately discuss the limitations of the retrospective design used in the intervention. Of note too is the racial/ethnic profile of the sample. How may this limit generalizability of the findings to a wider youth population? Provide further details on how the youth were selected for the intervention.
However, given the purpose of the study - which was to evaluate the FNLM outcomes, the discussion should have included a comparison/contrast with other such interventions
Author Response
Comment 1: Authors appropriately discuss the limitations of the retrospective design used in the intervention. Of note too is the racial/ethnic profile of the sample. How may this limit generalizability of the findings to a wider youth population? Provide further details on how the youth were selected for the intervention.
Thank you for these extremely insightful comments. We have expanded the discussion section to caution the reader on generalizing study findings to all adolescents as some subgroups (gender and race/ethnicity) had small sample sizes. Also, we provided futher details about how youth were selected for this intervention in the background:
“School staff and the FNLM adult staff (advisors and coordinators) recruit high school students (grades 10-12) who they think will be a good mentor. FNML staff are encouraged to consider recruiting youth who might not be involved in other activities but who have a lot to offer, and consider selecting mentors similare to the protégé population (e.g., gender, ethnicity, life interests, and whose strengths align with the needs ot the protégés). Youth can also self-refer to participate. Youth who want to be a mentor must complete an application, have parent’s/guardian’s consent, and participate in an oral interview with the advisor and program coordinator. Once they have been selected, they are required to sign a contract related to their commitment to the program. This includes their agreement to act as a role model at all times, make a commitment to spend at up to three hours a week dedicated to the mentoring program, and maintain a grade point average (GPA) of 2.5. There must be a three-year grade difference between mentors and protégés.”
Comment 2: However, given the purpose of the study - which was to evaluate the FNLM outcomes, the discussion should have included a comparison/contrast with other such interventions
Thank you for this comment. We have expanded the discussion to include current literature for comparative purposes and have added the corresponding citations.
“There is a vast body of literature showing that connecting youth with caring adult mentors supports positive outcomes [34-35]; however, there is much less research on the effectiveness of peer mentors. Studies, from across the globe, have found that peer-led substance use prevention programs are an effective approach to reducing substance use especially when there is careful attention to the selection and training of peer leaders [36-38]. While they are not 1:1 mentors, peer leaders serve as role models who are effective at changing social norms, providing alternatives to substance use, and in teaching ATOD refusal skills. Additional research shows that a strong positive peer network contributes to improved social and academic outcomes [39]. The current study contributes to a growing body of research demonstrating that peer mentoring of high school students with younger students benefits both the protégé [40] and the mentor [41].”
Again we are grateful to all of the reviewers for their thoughtful and constructive review of our manuscript. These comments helped improve the quality of our paper.